# Translational Applications of Wearable Sensors in Education: Implementation and Efficacy

**DOI:** 10.3390/s22041675

**Published:** 2022-02-21

**Authors:** Brendon Ferrier, Jim Lee, Alex Mbuli, Daniel A. James

**Affiliations:** 1School of Applied Sciences, Edinburgh Napier University, Edinburgh EH11 4BN, UK; a.mbuli@napier.ac.uk; 2Physiolytics Laboratory, School of Psychological and Clinical Sciences, Charles Darwin University, Darwin, NT 0909, Australia; jim@qsportstechnology.com; 3SABEL Labs, College of Health and Human Sciences, Charles Darwin University, Darwin, NT 0909, Australia

**Keywords:** disruptive innovation, wearable technology, STEM education

## Abstract

Background: Adding new approaches to teaching curriculums can be both expensive and complex to learn. The aim of this research was to gain insight into students’ literacy and confidence in learning sports science with new wearable technologies, specifically a novel program known as STEMfit. Methods: A three-phase design was carried out, with 36 students participating and exposed to wearable devices and associated software. This was to determine whether the technology hardware (phase one) and associated software (phase two) were used in a positive way that demonstrated user confidence. Results: Hardware included choosing a scalable wearable device that worked in conjunction with familiar and readily available software (Microsoft Excel) that extracted data through VBA coding. This allowed for students to experience and provide survey feedback on the usability and confidence gained when interacting with the STEMfit program. Outcomes indicated strong acceptance of the program, with high levels of motivation, resulting in a positive uptake of wearable technology as a teaching tool by students. The initial finding of this study offers an opportunity to further test the STEMfit program on other student cohorts as well as testing the scalability of the system into other year groups at the university level.

## 1. Introduction

The application and use of wearable technologies, such as inertial measurement units (IMUs), within sport and exercise science have become widespread over the past decade [1]. The implementation of IMUs for biomechanical analysis of sports has become a growing area in research and performance monitoring [1]. The advances in this new technology, along with increases in computing power, have enabled the continual development of IMUs for quantifying human movement.

### 1.1. Wearable Trends in Sport and Leisure

In recent years, wearables have seen enormous growth within the sporting and leisure domains [2]. Early in its development cycle, wearables were used progressively across medical and elite sporting domains as a means to collect data in more natural environments. One example is the monitoring of heart function through measuring blood pressure or using ECG within a medical or sports laboratory setting. These clinical forms of measurement offered only snapshots of information [3], whereas the use of wearables and the ability to measure these variables within the individual’s daily living or training environment has allowed greater insights into health or sports to be developed.

Facilitating this, the development of applied wearables was able to ‘ride the waves’ of technology trends. Accelerometers are an example of an integral component in today’s wearables. Accelerometers measure changes in inertia, and so when applied to the body, enable much of its biomechanics to be measured to a comparable level of accuracy to laboratory grade equipment [4]. Accelerometers had their birth in large-scale navigational equipment, but had their first wave of miniaturization as sensors for car airbags, which brought them down to MEMS-sized devices [5]. Then, with the advent of smart phones, they were incorporated as tilt sensors to determine screen orientation. This miniaturization roughly follows Moore’s law, which states that the number of transistors on a given area of silicon will double every 1.5 years [6]. Obviously for wearables, size is a critical component of uptake within a sport and leisure setting, with miniaturization allowing a corresponding reduction in power requirements, with battery size being a key component in the overall wearable dimensions.

Beyond the wearable nature of the sensors themselves, their associated support systems for data acquisition, storage, display, processing, and communication to the end user have been greatly facilitated by the trend of convergence, with many of today’s electronic devices essentially having the same components within them [7]. This has allowed wearables to appear in other socially acceptable and useful technologies, further increasing acceptance and utility within sport and leisure. These include watch and mobile phones, with mobile phones being such that the adherence to carrying them could reasonably allow these to be considered as wearable (despite their size).

In addition to the sensor technology, network connectivity of smart devices has allowed for the seamless connection of wearable data to end-user platforms, including analytics, social media and user-friendly display. This has driven a growing sophistication and desire for data in end users, even in commercial-grade products. As a result, harnessing this widespread availability of relatively cheap devices that consumers are hungry for makes it a potentially advantageous tool for education, in particular STEM education, where the use of these devices, along with the data collected, can be harnessed to improve educational engagement and outcomes.

### 1.2. Review of Technology Utilization within Education

The portability and accessibility of the IMU mean that this is a technology that has the potential be easily introduced into the teaching environment, with the adoption of wearables into the field of education sitting across several disciplinary boundaries. It is here, at the intersection of the technologists, educator, and the particular educational field that the translation of a successful technology has had a large, possibly disruptive effect for educators [8]. The use of technology through technologically enhanced learning has been defined as the ability to learn within an environment that has been enriched through the integration of digital technology [9]. This integration can include hardware devices such as laptops, mobile phones, and televisions, but can also include the use of wearables. Wearable technology has been utilized within a broad educational setting, enabling students to relate directly to the data presented [10]. Therefore, based on the user’s movement and the data presented within a practical class setting, the student can conceptualize the information and interpret the data presented [10]. This enables what was performed within a physical education class to be a topic addressed within mathematics, physics or biomechanics classes and also provides a broad educational experience for the student, possibly enhancing the student’s understanding.

Studies have shown that the integration of technology into the classroom offers possibilities of new approaches to teaching and learning, with computer software able to assist problem solving and allow students to explore concepts [11]. Examples within higher education indicate that there is a drive to use new technologies to engage the student and enhance their productivity [12]. Anecdotally, students new to the field of sport and exercise science tend to struggle with the concepts of angular velocity and acceleration and their practical application, and the use of wearables and the associated analysis of these concepts could potentially help these students to conceptualize these and other variables. Though not specifically documented in the field of sport and exercise science, the struggle to conceptualize and visualize abstract theory has been reported as an issue in other scientific fields such as physics [13]. Introductory courses are essential for underpinning knowledge in any science with material being taught in the form of lectures which do not foster an environment of active learning [13]. By introducing and allowing the students to engage actively with technology, this should allow students to see the concepts in action and relate the data gathered directly to specific movements they have performed.

Students use several technologies in their day-to-day lives and are comfortable with technology, but new technologies should be introduced in ways that make them accessible to everyone, with opportunities to train if necessary [14]. Ensuring the content and context of new technology that is introduced to students is important, including consideration of the appropriateness of the technology for the situation and learning outcomes of the session. When the technology is suited to the task, and the task is developed around student abilities, then there is an increase in student engagement; and if the students’ interest in technology can be stimulated, this can lead to increased learning [15]. This suggests that if the task is suitable and related to a sport or exercise that the student is familiar with, the introduction of wearable sensors in an educational environment should allow the student to engage and feel confident in learning how to use such technology. The aim of this study was to try to gain insight into students’ acceptance of and confidence in using a new wearable technology to learn basic concepts in a sport and exercise science setting.

## 2. Materials and Methods

The present study required a three-phase process investigating the design of the wearable technology appropriate for use within an educational setting, the development of an appropriate software to support the pedagogical application of wearable technology in a tertiary educational setting, and finally the assessment of the end user’s confidence in and acceptance of wearable technologies in understanding biomechanical concepts.

### 2.1. Wearable Technology Design

The development of wearable sensors for the monitoring of athletes has seen accelerated development since the early 2000s, when the sensors were developed in a kind of arms race towards various Olympic games. One of these, in particular, was the sport of swimming, with Australia [4] and the United Kingdom [16] seeing development as a competition. Since that time, swimming has been the subject of a number of studies into the use of wearables [17,18,19]. Swimming, in many ways, represents a kind of pinnacle of testing for groups implementing wearable sensors. An aquatic environment is harsh, with the smallest amount of drag being created having a negative effect on the adoption of wearable sensors within this sport. Swimming is well understood biomechanically and being largely linear in nature, it gives rise to a large number of metrics that the sport has adopted, such as race and split times, stroke counts and stroke length. These metrics are somewhat labor intensive to record, and require a number of different tools to quantify. Therefore, swimming is an ideal candidate for the automation provided when utilizing wearable sensors.

One of the greatest challenges in swimming and other sports is communicating and adoption by a largely non-technical audience. In this manner, many of the developed metrics, software and tools are required to be user friendly for their use and interpretation of data, as well as being relatable to the athlete [20]. This was therefore a natural transition for the translation of current available tools into an educational context.

Many other sports today also utilize wearable sensors, giving a large pool of applications to engage people, as well as apply within an educational setting. The utilization of wearables, as previously outlined, allows individuals to emotionally connect with the technology and product due to personal interest [10], which is seen as key to the adoption of new technologies [21]. Sports using wearables include snowboarding [20], athletics and the biomechanics of running [22], cross-country skiing [23] and team-based sports [24]. Of these, running, walking, and jumping, which are the primary means of locomotion and enablers for many physical activities, were seen as the most natural ‘first base’ for technology translation into an educational tool such as the STEMfit software package [25]. Briefly, the STEMfit concept evolved from a related project measuring physical literacy in school children. During this, it was quite apparent that many children had a disconnect with classroom activities, especially STEM-based subjects, but had a keen interest in learning about themselves. From these observations, we questioned whether combining self-interest, along with inherent interest in smart devices, and learning STEM could be possible. We decided, instead of developing complex technology or using high-end technology, to develop an end-user product that was readily available to the majority of teachers and students, i.e., wearable hardware and Excel-based software [8].

Thus, from amongst a wider variety of available wearable technologies, from consumer-grade wearables, where raw data access was not possible, through to high-end specialist devices, we decided to use middle-of-the-road technology that was as easy to use as a USB stick, yet provided access to raw sampled data as the most appropriate and considered it to match the needs of the learning environment, stakeholder expertise and available technologies [8]. The authors regard this as a beachhead to more sophisticated technologies in the future [26]. Recently, we have scaled the concept to test efficacy in higher education environments. At this point, curriculum development means the software is not freely available. However, future plans for curriculum-designed products that include the STEMfit software will be available for uptake by learning institutions. An open-source version of the software is also being considered to run along side the teaching package.

### 2.2. Development of the STEMfit Software Programme

The development of any product, in particular technology, can come from the domain of the technology (as a technology push) or that of the intended end-user group (as a demand push) [27]. There is often a large skill, interest and domain gap between these two groups. Therefore, it becomes quite a challenge for the developers of leading wearable technology sensors to develop something for school students as an educational tool. One approach to this is to conduct a customer-driven orientation, starting with a needs analysis of the student and various stakeholders [8]; we found that the technological literacy of the end users and availability of technologies are key considerations. This need to be considered along with optimization and sophistication of the technology. For students, as end users, the computational environments available to them are less likely to be specialized high-performance environments (e.g., Matlab availability and understanding), and this needs to be considered. Further, resource availability in schools and university departments may be limited to comparatively cheap devices if they are to be utilized in high volumes, impacting on the consideration and design of wearable technologies and the associated software supporting the devices. Easy integration of the technologies is a vital component in the standard operating environment (SOE) of corporatized computing environments, where custom devices and software are unlikely to have special drives and software to be easily installed and maintained.

### 2.3. Acceptance and Confidence Using the Technology

#### 2.3.1. Participants

Thirty-six (12 female and 24 male) first-year undergraduate Sports and Exercise Science students participated in two separate sessions (quantitative data capture and qualitative data collection) with 72 h between the two sessions. All participants were provided with a clear explanation of this study, and were asked to provide written informed consent before participation. This research and procedures were approved by the University Ethics Committee of Edinburgh Napier University (approval number: SAS0080) and in regulation with the Declaration of Helsinki for Medical Research involving human participants.

#### 2.3.2. Quantitative Data Capture Session

During the quantitative data capture session, 10 randomly allocated students were fitted with a single IMU (Human Activity Monitor (HAM-x16) Gulf Coast Data Concepts, USA) to help with the collection of the data, and illustrate the ease of use of the single IMU in the collection of kinematic data. Each HAM-x16 IMU contains a single InvenSense MPU-9250 9 axis sensor, which includes a triaxial accelerometer, a gyroscope and a magnetometer, as well as a digital motion processor to allow for an orientation solution [28]. Each sensor measured 56.1 × 39.4 × 15.2 mm and were set at a frame rate of 200 Hz. With the acquisition of the data being carried out using the HAM-x16 IMU, the sensors were orientated so that the x, y and z axes represented the longitudinal, anteroposterior and mediolateral axes of rotation when the participant was in an upright position.

During the quantitative data capture session, each participant was fitted with a single IMU to their back, in line with the posterior-superior-iliac-spine with the supplied elastic belt. This sacral location is both a convenient attachment point using a waist strap, and well supported in the literature for capturing gait biomechanics when compared to other locations [29]. Once fitted, each participant was instructed to perform a series of jumping tasks during the class, with the standing long jump test being used for data collection due to ease of analysis and confirmation of results with a standard tape measure. Prior to the performance of the standing long jump, the instructor started the sensor and instructed the performer to maximally jump for distance. Upon landing, the instructor informed the student to stand in place whilst the sensor was stopped, with the data stored internally on the IMU. All students were instructed upon the use of the IMU, with those not performing the jumps instructed to watch how the IMU can be fitted and implemented within the collection of the athlete performance.

After the jump session was completed, each IMU was removed from the participant for data extraction and file preparation from the instructor for the subsequent qualitative data collection session performed within the university computer laboratory. Prior to the qualitative data collection session, each file was trimmed to just contain the jump data to simplify the file and allow the students to maximize their use of the custom STEMfit software package.

#### 2.3.3. Qualitative Data Collection

Prior to the qualitative data collection session, one of the researchers reminded the class of the project aims, re-reading the participant information sheet and informing the students of the purpose of this study. All students were encouraged to ask questions and informed that their participation in this study had no impact on the running of the class session. Once informed, all students were provided an informed consent form, and those students who gave consent were provided with a questionnaire with questions adapted from the Intrinsic Motivation Inventory (IMI) [30] and the Technological Acceptance Model (TAM) section of the questionnaire devised by Chen [12] (see Appendix A). Once all the administrative work had been completed, the students were introduced to the custom STEMfit program, and the data collected at the previous session. Students were provided a written guide on the use of the STEMfit program to allow them to refer to it during the class. The students were then asked to analyze ten separate jumping files and provide a report based on the ten jumping files, presenting the data as if they were sports scientists reporting to a coach.

Upon completion of the session, each of the consenting students were then asked to complete the 14 separate questions within the questionnaire (Appendix A), utilizing the Likert-scale responses related to the experience using the STEMfit program. The questionnaire was based on 3 main themes from the IMI, interest/enjoyment, perceived motivation and pressure tension; and three main themes from the TAM, perceived usefulness, perceived ease of use and intention to use in the future.

Upon completion of the questionnaire, the students’ results were then tabulated and analyzed utilizing descriptive statistics reporting the mean and standard deviation under the 6 main themes and 14 sub-themes using GraphPad Prism version 9.1.1 for Windows (GraphPad Software, La Jolla California, CA, USA).

## 3. Results

### 3.1. Technology Design

When developing the technology, we needed a wearable product which required three main components: sensor system, data analysis suite and an educational program. Initially, SABEL Sense was used; however, proprietary drivers and complexity with data processing and analysis, particularly for younger students, was deemed an issue. Further, as the developed program was scaled up, we needed access to faster-moving and more responsive volume quantities that enabled straight forward and ease of use to adapt for an educational setting. We elected for a USB device that acts as a USB memory stick, allowing for on-board storage, resulting in the HAM-x16 IMU.

### 3.2. Development of the STEMfit Software Package

Regarding the educational client, we chose something available on all computing systems within an educational environment—Microsoft Excel. This decision allowed for the access of relatively sophisticated computing through VBA (Visual Basic for Applications), which was dynamically customizable for individual educational programming, as well as to a user ‘sandbox’ for data analysis using techniques familiar to most students and their tutors and teachers (Figure 1). This combination allowed vertical scaling from point and click analysis using VBA, through to user-driven customization through calculation and charting capabilities in spreadsheet programs.

### 3.3. Acceptance and Confidence UsingTechnology

From the results presented in Table 1 related to the IMI, the results were further separated into three separate factors to help identify elements associated with the students’ intrinsic motivation when being introduced to the use of the STEMfit software. Using the three separate factors—(a) interest/enjoyment; (b) perceived competence; and (c) pressure tension—the results illustrate the range of mean scores associated with the students’ intrinsic motivation.

In Table 1, the pressure tension mean scores ranged from 2.75 to 3.78, *‘I felt relaxed while doing the tasks’* (3.78, SD 0.83). In contrast, students scored enjoyment, *‘While I was working on the task, I was thinking about how much I enjoyed it’,* as the lowest on the 5-point Likert scale of the IMI (2.75, SD 0.77). Interestingly, when looking at the other responses associated with interest and enjoyment, the students gave the second highest score (3.61, SD 0.73) for *‘I found the task provided very interesting’*, indicating that they may not have enjoyed the task but the session provided some interest at that time.

When looking at the students’ acceptance of using the STEMfit software during the lesson, the results are further separated into three sections: (a) perceived usefulness; (b) perceived ease of use; and (c) behavioral intention to use (Table 2). Table 2 illustrates the various mean scores of the technical acceptance survey, with the results suggesting that the students’ acceptance levels during the class are quite similar, with the mean ranging from 2.78 to 3.39. The highest scores were recorded for the items *‘I like using software packages like Microsoft Excel’* within the *‘perceived usefulness’* section, with a mean score of 3.39 (SD 0.96); and *‘I am good at using computers’* within the *‘perceived ease of use’* section, with a mean score of 3.39 (SD 0.99).

Interestingly, when looking at the students’ responses to the use of the STEMfit program (Table 2), the students scored the software program highly as a tool for learning within a classroom setting, increasing their understanding (3.36, SD 0.76). However, when it came to using the STEMfit program outside of the classroom setting, the student cohort did not feel confident, with the lowest response under the *‘behavioral intention to use’* section, with a mean score of 2.78 (SD 1.05).

## 4. Discussion

The purpose of the present study was to gain insight into tertiary level students’ literacy and confidence in learning sports science concepts using new wearable technologies. This study was performed in three phases—design of the technology, development of the software package used within the educational setting, and students’ acceptance of and confidence in using the technology. From the results, it can be suggested that the design and development of the sensor and supporting software resulted in mixed responses to the assessment of motivation when using (Table 1) the new technology (Table 2) and in terms of acceptance among students.

### 4.1. Technology Design

When considering the design of the technology, the exponential development and penetration of wearable sensors into a variety of market sites needs to be considered. This was essential as there are now a greater range of wearable technologies among people within some community users around the globe [2]. This increase in the use and drive for wearables and other smart devices has penetrated previously conservative industries [7] and has become an enabler within new industries.

The application of wearable technology within an educational setting is rising [26,31], increasing both students’ and educators’ appetite for technology and data. This use of wearable technology within the educational setting has also been observed to be of increased intrinsic interest in physical activity [26,31].

During the development of the technology, the research team considered the sensor system required, the data analysis suite available for the implementation of the software package and the educational setting the sensors will be implemented within. As a result, the research team decided upon a USB device in the guise of the HAM-x16 IMU. The decision was based on the user-friendly nature of the device. Its simplicity of a “plug and play” approach, simple data logging process, and a direct download of a csv file enables users to interact with the device without the complexities of background running software such as MATLAB. Therefore, it strips away layers of technological requirements in order for students learning the use of wearables to just focus on the sport science such as kinematic outputs. While it may appear that there is no novelty in the technology used, it should be considered as stepping back to progress forward, allowing students with little or no affinity to technology to better engage—something that is desired by industry and governments worldwide. This does not detract from the importance or the place in teaching and learning of more technical software or hardware. Therefore, it possibly allows a steppingstone for students in this discipline to be confident in the fundamentals of wearable technology prior to diving deeper into using more complex systems.

From the findings, the research team’s decision to introduce the HAM-x16 IMU as the incorporated wearable sensor shows that it is a promising tool in the toolbox for any educator looking to increase engagement as well as educational outcomes within STEM. Next, the team focused upon the development of an appropriate software package.

### 4.2. Development of the STEMfit Software Package

As the aim of this project was to investigate the use of wearable technologies in learning biomechanical concepts, the team settled on a program design that utilized software readily available within an educational setting. This ensured the students introduced to the technology and software would have some recognition of the platform the STEMfit software package was based on.

This decision to utilize Microsoft Excel and the available VBA allowed for the development of a suitable wearable technology product for the education market, through the utilization of a multidisciplinary approach. When investigating the best platform to base the software package on, we had to leave behind the technologist’s ideals of seeking the most recent and cutting-edge technologies. Instead, the focus was on greater ease of use and lower cost of adoption in terms of price as well as training and technological literacy required within an educational setting for both educators and students. The intervention and the STEMfit program were designed for a very specific cohort of students, which is something of a beachhead [32] from which the researchers hope to incorporate more wearables which appeal to a wider range of topics. Furthermore, this approach enables scalability, where secondary and primary school environments can also use the program with software that is typically found in school computer systems, e.g., Excel. Therefore, usability of hardware becomes somewhat easier through specifically designed software that enables data to change into a format that is suited for STEM-based teaching and learning environments. As the STEMfit concept evolves, access to a greater range of technologies will be developed into the design.

### 4.3. Acceptance and Confidence UsingTechnology

From the results presented in Table 1 and Table 2, there appears to be strong acceptance of the technology among participants surveyed. When looking at the levels of motivation among the students during the task (Table 1) according to the 5-point Likert scale, we can see high to extremely high motivation among the class participants. This high level of intrinsic motivation could be seen because of the environment in which the intervention was occurring [30]. Ryan [30] outlined that if the environment where the individual is performing enhances the individual’s feelings of confidence, then their intrinsic motivation will be seen to increase. Interestingly, Ryan [30] further outlined that when students illustrate a high level of intrinsic motivation during an event, it can often indicate high levels of competence. Later studies looking at the use of technology within an educational setting agree with these findings, suggesting that an increase in intrinsic motivation illustrates an increase in the learning experience and motivational drive to learn [11,12]. This further indicates that the participants in the present study feel that the use of the STEMfit program in supporting their understanding of biomechanical concepts allowed them to feel confident and competent within their understanding during the session.

When looking at access to STEM, the increase in motivation illustrated by the participant group (Table 1) hints to the prospective interest and confidence in the pursuit and continuation of interest within the STEM environment [33]. Drazen [33] further explains that to see an increase in the uptake of STEM, the individual requires some aspect of self-efficacy and understanding of the topic. The aim of the STEMfit program is to engage students through interest with a student-centric approach [34]. This approach can be deemed a success from the responses in Table 1, with students indicating levels of motivation within the high to very high levels on a 5-point Likert scale.

When we looked closer at the TAM results in Table 2, it was also clear that there was high acceptance of the technology among participants. One explanation for the results may be the age group, which is generally representative of first-year undergraduate students, a specific age demographic (Z-generation) who typically take up new technologies [35]. This age group has grown up experiencing digital technology as part of their everyday lives [35]. Therefore, we would expect the students to typically accept the STEMfit program and be comfortable with using it in familiar surroundings.

When looking further at the individual components of the TAM results, it was reported in an earlier paper by Briz-Ponce and co-authors [36] that the results related to perceived usefulness provided an important predictor for the respondent’s attitude towards the use of technology. From the results in Table 2, we can see that that the overall score for perceived usefulness scored the highest rank overall, with the participants suggesting the use of the Microsoft Excel-based STEMfit program helped with their understanding. Further to this, it would seem the results suggest that the students perceived the session utilizing the STEMfit program as a positive experience; however, they have a moderate willingness to adopt the use of the program outside of the educational setting. The students’ confidence in their ability to use the software outside the classroom was the lowest score in the TAM (Table 2), suggesting that though students did feel confident in using the technology, supervision and guidance should be made available to them to enhance their learning prior to independent use outside the classroom. Furthermore, it is one of the prime objectives for STEMfit to better equip and ready students outside the classroom for STEM-related knowledge. The translation of wearable technology into the educational domain is intended to expose students to learning environments with resources found in everyday life. Since this project is in its early stages, the ongoing development of teaching curricula that encompass wearable technology as a tool to improve learning capacity and performance, enabling redesign and modification of curricula, will be the long-term aim of this project, with performance measures based on student knowledge and reflective feedback as well as uptake of technologies. For example, as students feel more comfortable with technology use, deeper learning into technology and more complex data extraction than is typical in sport science programs can be taught.

This feeling of being less comfortable using the program in unfamiliar environments may be more to do with the measure of self-confidence of the students, rather than the capacity to use the technology itself. Typically, when confronted with something outside their normal routine, e.g., an exam, young adults will undertake problem-focused coping strategies with the extreme being (in the exam scenario) not to sit the exam [37]. Being faced with the prospect of using STEMfit outside the relative security of a teacher-guided class environment may impact on confidence and self-efficacy.

## 5. Conclusions

Education is an expensive business; however, using cost-effective wearables has allowed engagement to be increased together with better learning outcomes. The development of wearables rides several worldwide trends, allowing them to be customized for education at a comparatively low cost whilst delivering a potential ‘disruptive intervention’. That is, with some out-of-the-box thinking, it is cheaper than existing programs, with low overheads for the educator. This is reflected in the first and second phases, where hardware was sought that is readily available and relatively simple to use. In conjunction with this, the use of readily available software (Excel) allows straight forward access for users to download files directly from the wearable devices. This approach is supported by outcomes in phase three indicating largely positive uptake and acceptance among students. This allows for future studies to monitor other student cohorts as well as test scalability into different year groups at the university level among those undertaking biomechanics.

## Figures and Tables

**Figure 1 sensors-22-01675-f001:**
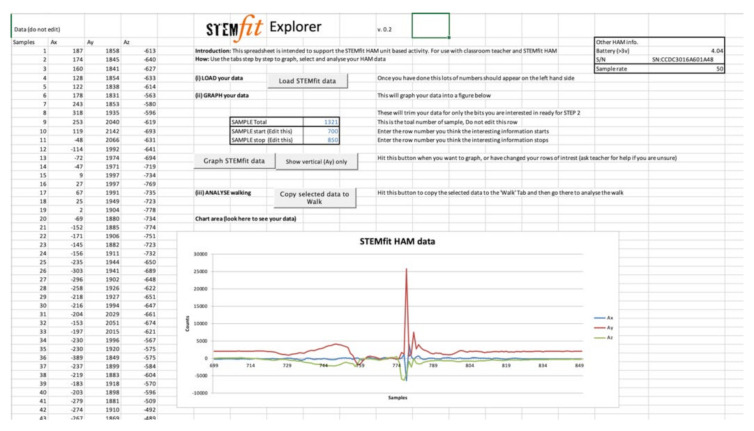
A sample of a typical Excel-based STEMfit interface.

**Table 1 sensors-22-01675-t001:** Intrinsic Motivation Inventory (IMI) (*n* = 36).

Factors	Mean	SD
**Interest/enjoyment**	**3.21**	**0.62**
Doing the task was fun	3.33	0.86
I found the task provided very interesting	3.61	0.73
While I was working on the task, I was thinking about how much I enjoyed it	2.75	0.77
**Perceived competence**	**3.01**	**0.85**
I felt pretty skilled at this task	2.89	0.95
I think I am pretty good at the task at hand	3.14	0.90
**Pressure tension**	**3.78**	**0.83**
I felt relaxed while doing the task	3.78	0.83

Notes: Variables in bold relate to the subscales within the Intrinsic Motivation Inventory. With each item and associated score presented under subscale.

**Table 2 sensors-22-01675-t002:** Technological Acceptance Model (TAM) (*n* = 36).

Factors	Mean	SD
**Perceived usefulness**	**3.32**	**0.66**
I like using software packages like Microsoft Excel	3.39	0.96
Using the STEMfit software as a tool for learning in a classroom setting increased my learning and academic understanding	3.36	0.76
Use of the STEMfit software as a tool for learning in a classroom setting increased my self-efficacy	3.22	0.72
**Perceived ease of use**	**3.22**	**0.57**
I am good at using computers	3.39	0.99
It is easy to use the STEMfit software as a tool for learning	3.08	0.91
My learning and understanding turned out to be easier for me by using the STEMfit software	3.19	0.85
**Behavioral intention to use**	**2.96**	**0.69**
I would feel confident using the STEMfit software outside of the classroom to increase my learning and academic understanding	2.78	1.05
I am more confident in my understanding of the applications of micro technology in the learning of biomechanical principles	3.14	0.83

Notes: Variables in bold relate to the subscales within the Intrinsic Motivation Inventory. With each item and associated score presented under subscale.

## Data Availability

The data presented in this study are available on request from the corresponding author.

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
