# Peer review of "Translational Applications of Wearable Sensors in Education: Implementation and Efficacy"

_sensors, 2022, doi:10.3390/s22041675_

Round 1

Reviewer 1 Report

In the manuscript “Translational applications of wearable sensors in education: Implementation and efficacy”, the authors investigated the user experiences when the wearable motor devices, including the inertial measurement unit (IMU), are utilized for educational purposes. In this work, the direction of device applications is well demonstrated for the student consumers. Generally, it could have been conceptually common that a wearable device will provide an abundant educational experience in a related college department (e.g., sports engineering). Based on these concept and hypothesis, it is very encouraging that the educational experience is quantitatively measured in this result through appropriate experiments and questionnaires.

However, from the reviewer’s point of view, this work is considered to be inconsistent with the aims and scopes of Sensors. Except for the interesting research motivation, the major weakness in my opinion is that the detailed experimental contents are insufficiently provided. The following parts show some deficiencies of this manuscript:

  1. The information on the experiments and processes (i.e., graphs and figures) on the results, including STEMfit, is insufficiently provided. It is confirmed that the result from the survey of the subjects was solely suggested as the meaningful results. In this regard, non-specialists from broader fields will not understand how the experiment was performed on the instrument (i.e., wearable devices) as well as the sensors such as IMU.
  2. Considering the psychological changes of students, it is thought that a journal in the applied medical fields or statistical fields would be more suitable for this study.

Thus, the manuscript is not well suited for the publication in Sensors. I recommend the authors either submit to a specialized journal or make the work accessible to a broader audience.

Author Response

Reviewers Comments

Reviewer One

Open Review

English language and style

( ) Extensive editing of English language and style required
( ) Moderate English changes required
(x) English language and style are fine/minor spell check required
( ) I don't feel qualified to judge about the English language and style

Yes

Can be improved

Must be improved

Not applicable

Does the introduction provide sufficient background and include all relevant references?

(x)

( )

( )

( )

Is the research design appropriate?

( )

(x)

( )

( )

Are the methods adequately described?

( )

( )

(x)

( )

Are the results clearly presented?

( )

( )

(x)

( )

Are the conclusions supported by the results?

( )

( )

(x)

( )

Comments and Suggestions for Authors

In the manuscript “Translational applications of wearable sensors in education: Implementation and efficacy”, the authors investigated the user experiences when the wearable motor devices, including the inertial measurement unit (IMU), are utilized for educational purposes. In this work, the direction of device applications is well demonstrated for the student consumers. Generally, it could have been conceptually common that a wearable device will provide an abundant educational experience in a related college department (e.g., sports engineering). Based on these concept and hypothesis, it is very encouraging that the educational experience is quantitatively measured in this result through appropriate experiments and questionnaires.

However, from the reviewer’s point of view, this work is considered to be inconsistent with the aims and scopes of Sensors. Except for the interesting research motivation, the major weakness in my opinion is that the detailed experimental contents are insufficiently provided. The following parts show some deficiencies of this manuscript:

  1. The information on the experiments and processes (i.e., graphs and figures) on the results, including STEMfit, is insufficiently provided. It is confirmed that the result from the survey of the subjects was solely suggested as the meaningful results. In this regard, non-specialists from broader fields will not understand how the experiment was performed on the instrument (i.e., wearable devices) as well as the sensors such as IMU.

Response:

We have added further detail on the results to the text and added content to the development of the STEMfit software package (including figure one) enabling the reader to understand the interface and what the students were being asked to base their motivation and acceptance upon. We contend that measures around engagement are critical to the uptake of wearable technologies, adheres to their usage, and engagement with the technologies is critical for its adoption and entirely consistent with the objectives of the wearable technologies section “The Wearables section of Sensors publishes original peer-reviewed papers covering all aspects of wearable sensors, devices, and electronics, and their applications for humans, animals and livestock.”

  1. Considering the psychological changes of students, it is thought that a journal in the applied medical fields or statistical fields would be more suitable for this study.

Response:

The authors wish to thank the reviewer for this comment. However, the authors consider that the intent of the manuscript has been overlooked. The intent is to have the document accessible to a broader audience (as per the recommendation in the following paragraph). Already having a cross discipline team of authors goes towards this objective by knowing their respective areas of research. Furthermore, and in line with the high-level feedback at the top of this document, this manuscript fits into the journal’s objectives. Therefore, giving information to professionals within the sensor development field of the typical experiences of those applying technology. Meaning that genuine information is available to those who may (and often) can only assume or guess what user experiences are. Therefore, the information will be valuable to those who wish to develop wearable technology further.

Thus, the manuscript is not well suited for the publication in Sensors. I recommend the authors either submit to a specialized journal or make the work accessible to a broader audience

This statement appears to be inconsistent with the aims and objectives of the wearable technologies section of the journal “The Wearables section of Sensors publishes original peer-reviewed papers covering all aspects of wearable sensors, devices, and electronics, and their applications for humans, animals and livestock.”

Reviewer 2 Report

The article presents an application of wearables in education and investigates students’ literacy and confidence in learning sports science.

The article needs improvements in different parts. The major issue is that while the study has the merit, its contributions are weak.

- (4.2) the technology design part is not strong. There are many comfortable, powerful and affordable wearables in this area. The justification for choosing this device is not sufficient to be one of the article's contributions.

-(4.3) the development of software package also is weak. There are many modern and advanced software options today that have a very friendly and easy interface.

Also, the article didn't include any screenshots of the interface to understand how it looks like and how students interact with it.
If only the Excel was used, there is not much development involved to be considered as a contribution.

Also, please elaborate what type of data analysis was conducted and how results were visualised/presented.

Sections 1.1 and 1.2 are verbose at the moment and have some basic details that are not necessary. These sections can be shortened with important review of current, related works.

Section 1.2 needs adding more recent works in education utilising sensing and mobile devices and discussing what the state of the art is, and then link it to this paper study that should aim to enhance it or address its open issues.

The conclusion part do not specify the findings that can be considered as new knowledge or a solution to a certain research problem (with providing evidence from the literature).

- About participants and study instrument - the article mentions only 10 out of 36 were involved in the jumping activity to collect data.
Could you clarify how many were involved in the next phase, and what was the rationale behind considering these numbers for each phase of study.
Also please elaborate on how you selected and designed the instruments (and questions) using IMI and TAM, and why these two models were chosen. What questions were included (are these only the ones included in Table 1 and 2), if not, you could include them as an appendix.
- in Section 2.3.2, it mentions about the location of sensors "in line with the posterior-superior-iliac-spine with the supplied elastic belt", could you explain how the optimal location was decided.

The study only looked into the students' literacy and confidence in using wearables for learning Sports Science but the important aspect is missing which is how this approach can improve student learning and performance  (which should be the ultimate goal) and how you would measure it.

Author Response

Reviewer Two

Open Review

English language and style

( ) Extensive editing of English language and style required
( ) Moderate English changes required
(x) English language and style are fine/minor spell check required
( ) I don't feel qualified to judge about the English language and style

Yes

Can be improved

Must be improved

Not applicable

Does the introduction provide sufficient background and include all relevant references?

( )

( )

(x)

( )

Is the research design appropriate?

( )

( )

(x)

( )

Are the methods adequately described?

( )

( )

(x)

( )

Are the results clearly presented?

( )

( )

(x)

( )

Are the conclusions supported by the results?

( )

( )

(x)

( )

Comments and Suggestions for Authors

The article presents an application of wearables in education and investigates students’ literacy and confidence in learning sports science.

The article needs improvements in different parts. The major issue is that while the study has the merit, its contributions are weak.

(4.2) the technology design part is not strong. There are many comfortable, powerful and affordable wearables in this area. The justification for choosing this device is not sufficient to be one of the article's contributions.

Response:

The reviewers agree that there are powerful and affordable hardware available (some even developed by the authors…). However, the device was chosen for its simplicity of recording and extracting data. The focus of this paper is on the engagement aspects of using the technology, which is quite novel in the journal, and we think will provide readers with insight.

The technology design section has been significantly strengthened by adding a section in this regard

(4.3) the development of software package also is weak. There are many modern and advanced software options today that have a very friendly and easy interface.

Response:

The reviewers agree that there are advanced software packages available (again, some even developed by the authors). However, a main underlying principle with the research that this manuscript is part of is the point of making available to users, a product that is readily available. Many software packages either require complex software to run e.g., MATLAB. This type of software is expensive to purchase and even if it runs silently in the background, is quite often wanting processing power that many potential users have no capacity to provide. The aim of the overall project is to make an educational product that is available to all students, not just those that can afford it. Design has focused on an appropriate technology, so that programming and familiarity is not an additional educational barrier. Lowest common denominator technologies of spreadsheet usage and basic windows familiarity was used in this instance. While we have made this defence, we also see the importance to highlight this point in the manuscript. Therefore, a sentence has been added to rationalise our objective. It has been added to the paragraph commencing at line 336, with the sentence starting at line 345. It reads:

While it may appear that there is no novelty in the technology used, it should be considered as stepping back to progress forward, allowing students with little or no affinity to technology to better engage – something that is desired by industry and governments worldwide.”

Also, the article didn't include any screenshots of the interface to understand how it looks like and how students interact with it. If only the Excel was used, there is not much development involved to be considered as a contribution.

Response:

The contribution here is not the wearable technology per se, but how to make wearable technology accessible in an educational context, thus rather than a technology push, and demand-pull approach was used. Additional screen shots (Figure 1 in section 3.2) and explanation around the technology is added to the text to aid in this discourse. In this way this paper makes quite a novel contribution to a technically heavy journal that seeks to widen it remit to other aspects of the technology as given in the section charter “The Wearables section of Sensors publishes original peer-reviewed papers covering all aspects of wearable sensors, devices, and electronics, and their applications for humans, animals and livestock.”

Also, please elaborate what type of data analysis was conducted and how results were visualised/presented.

Response:

Descriptive statistics were utilised for this study, with the mean and standard deviation for all 6 main themes and 14 sub-themes represented in tables 1 and 2.

Sections 1.1 and 1.2 are verbose at the moment and have some basic details that are not necessary. These sections can be shortened with important review of current, related works.

Response:

The response from several reviewers indicates that more detail is required, it is difficult to be both shortened and provide basic details that are requested. 

Section 1.2 needs adding more recent works in education utilising sensing and mobile devices and discussing what the state of the art is, and then link it to this paper study that should aim to enhance it or address its open issues.

Response:

As asked by reviewer three, the authors have added content to provide justification for the content with section 1.2 that helps to expand and relate the study better. Content added in lines 97-104 reads.

“Though not specifically documented in the field of sport and exercise science, the struggle to conceptualize and visualize abstract theory has been reported as an issue in other scientific fields such as physics [13] . Introductory courses are essential for underpinning knowledge in any science with material being taught in the form of lectures which do not foster an environment of active learning [13]. By introducing and allowing the students to engage actively with technology it should allow to see the concepts in action and relate the data gathered directly to specific movements they have performed.”

The conclusion part do not specify the findings that can be considered as new knowledge or a solution to a certain research problem (with providing evidence from the literature).

Response:

The topic of the paper is the translational application of wearable technologies to another discipline. This activity is essential if we are to see the adoption of wearable technology continue to grow. The conclusions are around the challenges of such, and we think it would be of interest to readers who are interested in “all aspects of wearable technology” as provided in the charter for the section “The Wearables section of Sensors publishes original peer-reviewed papers covering all aspects of wearable sensors, devices, and electronics, and their applications for humans, animals and livestock”.

About participants and study instrument - the article mentions only 10 out of 36 were involved in the jumping activity to collect data. Could you clarify how many were involved in the next phase, and what was the rationale behind considering these numbers for each phase of study.

Response:

A convenience sample of 10 participants performed the jumping part of the study, as the focus was the class observe the data collection part of the study. The focus of the class and the study was around the data analysis and interpretation aspect of the session. Therefore, all 36 students participating in the class were involved in the use of the STEMfit software and its use in the analysis of the jump data.

Also please elaborate on how you selected and designed the instruments (and questions) using IMI and TAM, and why these two models were chosen. What questions were included (are these only the ones included in Table 1 and 2), if not, you could include them as an appendix.

Response:

The IMI and TAM was chosen and adapted to fit the aim of the study performed. The authors acknowledge that there are many other resources that we could have adopted. However, the study by Chen seemed to run parallel to the aims and purpose of the present study. Therefore, we felt it was a good fit to the current study. The adopted questionnaire has been included as an appendix.

In Section 2.3.2, it mentions about the location of sensors "in line with the posterior-superior-iliac-spine with the supplied elastic belt", could you explain how the optimal location was decided.

Response:

The use of wearable technology for biomechanical monitoring is well established, we include a historical well cited paper (+200 citations) in the text (Kavanagh et. al) that shows multiple sensors on the body capturing gait events. The authors chose the sacrum/PSIS as it is a well-recognised stable landmark that is close to the bodies centre of mass. This was the rationale for the choice of this placement position. (lines 217-219)

The study only looked into the students' literacy and confidence in using wearables for learning Sports Science, but the important aspect is missing which is how this approach can improve student learning and performance (which should be the ultimate goal) and how you would measure it.

Response:

We thank the reviewer for their insights, a section has been added to the discussion around supporting learning goals. This study was focused on the translation of wearable technology into the educational domain, engagement is a well-known precursor to learning gains.  To clarify this, additional information has been provided in the Discussion at the end of the paragraph that begins on line 406, with the extra information commencing on line 418. It reads:

“Furthermore, it is one of the prime objectives for STEMfit to better equip and ready students outside the classroom for STEM related knowledge. The translation of wearable technology into the educational domain is intended to expose students to learning environments with resources found in everyday life. With this project being in its early stages, the ongoing development of teaching curricula that encompasses wearable technology as a tool to improve learning capacity and performance. Reflective feedback from students will enable redesign and modification of curriculums will be the long term project aim with performance measure based on student knowledge and uptake of technologies as the basis of measurement of learning capabilities. For example, as students feel more comfortable with technology use, deeper learning into technology and more complex data extraction than is typical in sport science programmes can be taught.”

Reviewer 3 Report

Thank you for your research work focused on the introduction of STEM activities based on the use of digital wearable technologies in the teaching curriculum. I find that this work is very necessary at this time in which, in many countries because of the pandemic, the digital transformation of education has accelerated and significant investments are being made in digital technologies for the classroom.

I think your paper provides valuable information on critical aspects for the incorporation of wearable technology in classrooms. In addition, I have found it an interesting and well explained research work that can be considered a first pilot project that can inspire many teachers and technologists to collaborate in this area. Here are some minor revisions for the authors in order to improve your article for publication:

  • Lines 17 to 24: The format in which results and conclusions are included in the abstract are a bit strange. Please, integrate results and conclusions as connected sentences with the contents of the abstract.
  • Lines 70 to 71: This sentence must be rewritten to connect in a better way with the previous one.
  • Lines 93 to 97: This sentence is included in the technology review section, for this reason it is hoped that this information can be justified in some way, based either on the authors' own studies or on some literature reference.
  • Lines 105 to 112: This part of the article is hard to read. I suggest that the authors rewrite it to make it easier for future readers to understand.
  • Lines 114 to 119 and following subsections titles: In the introductory paragraph of section 2, the three levels of its methodology are identified by names that are not exactly the same used in the titles of subsections 2.1, 2.2 and 2.3. The homogeneity in the nomenclature facilitates a faster and easier reading for the user.
  • Line 169: An introductory paragraph to section 2.3 is missing. Include in this introduction a block diagram showing the different parts of its methodology and the main data of each stage (number of participants, type of hardware and software used, duration, dates on which each stage was carried out, etc.) it would be very useful to facilitate the understanding and replication of your methodology.
  • Line 192-193: Why did you choose the jump series task for your study? Is it based on a standard test or did you choose it for any reason that you can justify in the paper?
  • Line 212: You indicate that the questionnaires used were made by adapting IMI questions. If your questionnaire does not exactly follow a standard such as IMI, it is advisable to include it as an annex to your article.
  • Line 222: The same here, please provide the 14 questions in an annex.
  • Line 243: Please, explain what does VBA means the first time that this acronym is used in the paper.
  • Section 4: A general comment related to this section and also to the design of the questionnaires. As far as I can deduce from the content of your paper, the questionnaires used in your research only included closed questions in which the students had to answer with a Likert 5 scale. From my point of view, including some open questions in their questionnaires would have enriched the analysis of the results, allowing to obtain a richer feedback of the experiences of the users and valuable information to improve the design of your program. This could be an interesting point to be included in a Future Research section for your paper.
  • Line 293: What do you mean by “mixed responses in the motivation”, please explain it in a different way to clarify it.
  • Finally, this work, as I have commented at the beginning of this evaluation, can be a first pilot work to be followed by many interesting research works. For this reason, I think it would be of great interest to expose the main limitations of their research work and the future lines of research to which these limitations give rise. As an example, this work was conduct in a group of 36 students of the same university. This is a relevant limitation to generalize their results, and therefore proposing a future research line with several groups of students from different universities and countries can be relevant for advancing in the development of this line of research.

Author Response

Reviewer Three

Open Review

English language and style

( ) Extensive editing of English language and style required
( ) Moderate English changes required
( ) English language and style are fine/minor spell check required
(x) I don't feel qualified to judge about the English language and style

Yes

Can be improved

Must be improved

Not applicable

Does the introduction provide sufficient background and include all relevant references?

(x)

( )

( )

( )

Is the research design appropriate?

(x)

( )

( )

( )

Are the methods adequately described?

(x)

( )

( )

( )

Are the results clearly presented?

( )

(x)

( )

( )

Are the conclusions supported by the results?

(x)

( )

( )

( )

Comments and Suggestions for Authors

Thank you for your research work focused on the introduction of STEM activities based on the use of digital wearable technologies in the teaching curriculum. I find that this work is very necessary at this time in which, in many countries because of the pandemic, the digital transformation of education has accelerated and significant investments are being made in digital technologies for the classroom.

I think your paper provides valuable information on critical aspects for the incorporation of wearable technology in classrooms. In addition, I have found it an interesting and well explained research work that can be considered a first pilot project that can inspire many teachers and technologists to collaborate in this area. Here are some minor revisions for the authors in order to improve your article for publication:

We thank the reviewer for their responses as to the aims and objectives of the project, changes below are undertaken in the revised manuscript

  • Lines 17 to 24: The format in which results and conclusions are included in the abstract are a bit strange. Please, integrate results and conclusions as connected sentences with the contents of the abstract.

Response:

Authors have incorporated the text more and removed the section heading of conclusion. The amended section (Lines 17-24) now reads.

Results: Hardware included choosing a scalable wearable device that worked in conjunction with familiar and readily available software (Microsoft Excel) that extracted data through VBA coding. This allowed for students to experience and provide survey feedback on the usability and confidence gained when interacting with the STEMfit program. Outcomes indicated strong acceptance of the program with high levels of motivation. Resulting in a positive uptake of use of wearable technology as a teaching tool by students. This initial finding of the study offers an opportunity to further test the STEMfit program on other student cohorts as well as testing the scalability of the system into other university year levels

  • Lines 70 to 71: This sentence must be rewritten to connect in a better way with the previous one.

Response:

Sentence has been addressed and now reads “Where the use of these devices, along with the data collected can be harnessed to improve educational engagement and outcomes” (Lines 70-71)

  • Lines 93 to 97: This sentence is included in the technology review section, for this reason it is hoped that this information can be justified in some way, based either on the authors' own studies or on some literature reference.

Response:

Have added content to provide justification for the content in the sentence in lines 93-97. Content added in lines 97-104 reads;

“Though not specifically documented in the field of sport and exercise science, the struggle to conceptualize and visualize abstract theory has been reported as an issue in other scientific fields such as physics [13] . Introductory courses are essential for underpinning knowledge in any science with material being taught in the form of lectures which do not foster an environment of active learning [13]. By introducing and allowing the students to engage actively with technology it should allow to see the concepts in action and relate the data gathered directly to specific movements they have performed.”

  • Lines 105 to 112: This part of the article is hard to read. I suggest that the authors rewrite it to make it easier for future readers to understand.

Response:

Text in lines 105 to 112 have been rewritten to state “Suggesting that if the task is suitable and related to a sport or exercise related activity that the student is familiar with, the introduction of wearable sensors in an educational environment should allow the student to engage and feel confident in learning how to use such technology. The aim of the study was to try to gain an insight into student acceptance and confidence of using a new wearable technology to learn basic concepts in a sport and exercise science setting.” (Lines 111-117)

  • Lines 114 to 119 and following subsections titles: In the introductory paragraph of section 2, the three levels of its methodology are identified by names that are not exactly the same used in the titles of subsections 2.1, 2.2 and 2.3. The homogeneity in the nomenclature facilitates a faster and easier reading for the user.

Response:

Subsections have been re-named, but not in a manner to provide complete homogeneity. However, the authors feel that the changes do facilitate an understanding as to how the section titles relate to the text outlined in lines 126-131.

  • Line 169: An introductory paragraph to section 2.3 is missing. Include in this introduction a block diagram showing the different parts of its methodology and the main data of each stage (number of participants, type of hardware and software used, duration, dates on which each stage was carried out, etc.) it would be very useful to facilitate the understanding and replication of your methodology.

Response:

The authors thank you for this comment but feel that the information being asked for is included within sections 2.3.1 and 2.3.2. The text in section 2.3.1 clearly outlines 36 participants for the data analysis section. Section 2.3.2 clearly states 10 students during the data collection session. Section 2.3.2 clearly defines the hardware and software used throughout the study. Timings between sessions was also outlined within section 2.3.1, and the authors would like to ask about the reasoning for noting the duration of each session as never seen this in a methodology section, as well as the dates. The authors would like to ask how this information is relevant, and how does the inclusion of this information facilitate replication?

  • Line 192-193: Why did you choose the jump series task for your study? Is it based on a standard test or did you choose it for any reason that you can justify in the paper?

Response:

Comment addressed and sentence amended to outline testing rationale. The sentence now states “Once fitted, each participant was instructed to perform a series of jumping tasks during the class, with the standing long jump test being used for data collection due to ease of analysis and confirmation of results with a standard tape measure” (See lines 218-220)

  • Line 212: You indicate that the questionnaires used were made by adapting IMI questions. If your questionnaire does not exactly follow a standard such as IMI, it is advisable to include it as an annex to your article.

Response:

Questionnaire has been added as an Appendix at the end of the paper.

  • Line 222: The same here, please provide the 14 questions in an annex.

Response:

Questionnaire has been added as an Appendix at the end of the paper.

  • Line 243: Please, explain what does VBA means the first time that this acronym is used in the paper.

Response:

The term VBA means Visual Basic for Applications, and has been defined upon first use (Lines 272-273)

  • Section 4: A general comment related to this section and also to the design of the questionnaires. As far as I can deduct from the content of your paper, the questionnaires used in your research only included closed questions in which the students had to answer with a Likert 5 scale. From my point of view, including some open questions in their questionnaires would have enriched the analysis of the results, allowing to obtain richer feedback of the experiences of the users and valuable information to improve the design of your program. This could be an interesting point to be included in a Future Research section for your paper.

Response:

The authors would like to thank the reviewer for this observation and agree that some open ended or focus group aspect to the study design would have strengthened the results. However, time associated with the teaching was a hindrance to the overall study. The students were undertaking a module that composed of aspects of physiology, psychology and biomechanics, leaving little time to focus upon and expand on the students experience.

  • Line 293: What do you mean by “mixed responses in the motivation”, please explain it in a different way to clarify it.

Response:

The sentence has been amended to define and clarify meaning. The sentence now reads “From the results provided, it can be suggested that the design and development of the sensor and supporting software provided mixed responses within the respondent’s assessment of motivation when using (Table 1), and acceptance of the new technology (Table 2) introduced.” (Lines 324-327)

  • Finally, this work, as I have commented at the beginning of this evaluation, can be a first pilot work to be followed by many interesting research works. For this reason, I think it would be of great interest to expose the main limitations of their research work and the future lines of research to which these limitations give rise. As an example, this work was conduct in a group of 36 students of the same university. This is a relevant limitation to generalize their results, and therefore proposing a future research line with several groups of students from different universities and countries can be relevant for advancing in the development of this line of research.

Response:

Again, the authors would like to thank the reviewer for their comments. Overall, the comments have helped to focus the paper and we feel the additional information has strengthened the paper. Regarding this final comment, the authors feel that the final sentence of the paper (Lines 456-458) addresses this aspect as it details future studies and the use of other student cohorts and scalability.

Reviewer 4 Report

The importance of wearables in education is an emerging field and the authors have done a commendable job making headway into adding to much-needed research in this area.  A few comments will improve the paper and position it for greater impact.

The title has significant appear in the use of the word "wearables" and this was an attraction to reading further. Ultimately, though, the idea that only an IMU was the subject of investigation felt slightly misleading.  There are so many low-cost wearables on the market and the authors should at least consider providing a more compelling foundation for the IMU as indicative of what experience might be expected with any other wearable. It is, in deed, a HUGE challenge to integrate more than one sensor output together from commercially available sensors, and that subtlety (and, again, challenge) is often lost when discussing wearables.  Originally there was an expectation that data harmonization between sensors would be discussed.  Further, Lines 35-39 understate the major challenge in usability of data from different sensors. You hint at some of these challenges in the introduction, but it does not appear to be addressed in the body of the research. This erodes the implications of Line 83-84; ease of access to data from many wearables involves many steps and data formats very clumsy to use (without corporate subscription).

The use of the STEMfit software package seems entirely appropriate, but there is a self-reference to [24] (line 148) that begs for more detail: is the package opensource? Given the central role of the package in the research, a bit more insight on the selection and possible alternatives would enhance the value of the research to others.

The qualitative analysis of student results is a major component to the report and provides foundational information on potential research impact. We know that self-reporting of assessments has bias, so it would balance the paper and enhance research value to know what third-party assessment of the qualitative data was collected and analyzed.  For instance, did an instructor conduct interviews for cross-checking?  Were classroom observations conducted or planned? While having such data is very (very) difficult to obtain, it enhances the research to acknowledge the need.

This is an important research effort that deserves publication given the fundamental work and its application can be of great value to educators.

Author Response

Reviewer Four

Open Review

English language and style

( ) Extensive editing of English language and style required
(x) Moderate English changes required
( ) English language and style are fine/minor spell check required
( ) I don't feel qualified to judge about the English language and style

Yes

Can be improved

Must be improved

Not applicable

Does the introduction provide sufficient background and include all relevant references?

( )

( )

(x)

( )

Is the research design appropriate?

( )

(x)

( )

( )

Are the methods adequately described?

( )

(x)

( )

( )

Are the results clearly presented?

(x)

( )

( )

( )

Are the conclusions supported by the results?

( )

(x)

( )

( )

Comments and Suggestions for Authors

The importance of wearables in education is an emerging field and the authors have done a commendable job making headway into adding to much-needed research in this area.  A few comments will improve the paper and position it for greater impact.

The title has significant appear in the use of the word "wearables" and this was an attraction to reading further. Ultimately, though, the idea that only an IMU was the subject of investigation felt slightly misleading.  There are so many low-cost wearables on the market and the authors should at least consider providing a more compelling foundation for the IMU as indicative of what experience might be expected with any other wearable. It is, in deed, a HUGE challenge to integrate more than one sensor output together from commercially available sensors, and that subtlety (and, again, challenge) is often lost when discussing wearables.  Originally there was an expectation that data harmonization between sensors would be discussed. 

Response:

The focus in this paper was the translation of wearable technology to the educational domain. The chief challenges here are not to push the technology, rather instead to look at the demand-pull forces and use these. Thus, technology selection took an agnostic approach and examined the needs of the educational environment which is moe full described in a previously published paper referenced in the text. In the longer term it is envisaged that this provides a beachhead experience in the education domain to allow the translation of more sophisticated technologies. A paragraph in section  2.1 has been added in the text to further elaborate on this rationale for the reader.

“Thus, from amongst a wider variety of available wearable technologies from consumer grade wearables, where raw data access was not possible through to high end specialist devices and middle of the road technology that was as easy to use as a USB stick, yet provided access to raw sampled data was the most appropriate and matched the needs of the learning environment, stakeholder expertise and available technologies [8]. The authors regard this as a beachhead to more sophisticated technologies in the future [31]. “

Further, Lines 35-39 understate the major challenge in usability of data from different sensors. You hint at some of these challenges in the introduction, but it does not appear to be addressed in the body of the research. This erodes the implications of Line 83-84; ease of access to data from many wearables involves many steps and data formats very clumsy to use (without corporate subscription).

Response:

We appreciate the reviewers comments in this regard and have expanded the text here for clarity. The expansion of this has been made in section 4.2 of the Discussion (where it has already been partially addressed). This has been added to the paragraph commencing on line 358, with the comment starting at line 369. It reads:

Therefore usability of hardware becomes somewhat easier through specifically designed software that enables data to changed into a format that is suited for STEM based teaching and learning environments. As the STEMfit concept grows, access to a greater range of technologies will be developed into the design.”

The use of the STEMfit software package seems entirely appropriate, but there is a self-reference to [24] (line 148) that begs for more detail: is the package opensource? Given the central role of the package in the research, a bit more insight on the selection and possible alternatives would enhance the value of the research to others.

Response:

We thank the reviewer for their insightful comment.  Some expansion of historical work relating to the development of the package is included. Noting it is challenging to add detail and keep the paper to reasonable length and the focus to the topic at hand. The description can be found at the end of the relative paragraph (starting at line 145), with the addition commencing at line 153. It reads:

Briefly, the STEMfit concept evolved from a related project measuring physical literacy in school children. During this, it was quite apparent that many children had a disconnect with classroom activities, especially STEM based subjects, but had keen interest about themselves. From these observations, we questioned whether combining self interest, along with inherent interest in smart devices, and learning STEM could be possible. We decided that instead of developing complex and using high end technology, to develop an end user product that was readily available to the majority of teachers and students i.e. wearable based hardware, and Excel based software [8]. Recently we have scaled the concept to test the efficacy in Higher education environments. At this point curriculum development means the software is not freely available. However, future plans are for curriculum designed products that include the STEMfit software will be available for uptake by learning institutions. An open source version of the software is also being considered to run along side the teaching package.”

The qualitative analysis of student results is a major component to the report and provides foundational information on potential research impact. We know that self-reporting of assessments has bias, so it would balance the paper and enhance research value to know what third-party assessment of the qualitative data was collected and analyzed.  For instance, did an instructor conduct interviews for cross-checking?  Were classroom observations conducted or planned? While having such data is very (very) difficult to obtain, it enhances the research to acknowledge the need.

Response:

Two of the authors were on hand during both the data collection and data analysis sessions. No actual formal assessment was carried out or focus groups performed. However, the reviewers’ comments are greatly appreciated and will be considered in future studies looking at wearable technology integration into educational settings, as it is a valid observation.

This is an important research effort that deserves publication given the fundamental work and its application can be of great value to educators.

Response:

We thank the reviewer for their summative comment

Round 2

Reviewer 2 Report

Authors addressed all the comments to their best ability and the paper has improved.

The future work needs to use more modern technologies and more sophisticated data analysis. 

Reviewer 4 Report

Recent changes and additions and changes by the authors have improved the manuscript and addressed any issues.